# Adaptively Building a Video-language Model for Video Captioning and Retrieval without Massive Video Pretraining

Zihao Liu
Communication University of China
State Key Laboratory of Media Convergence and
Communication
Beijing, China
liuzihao@cuc.edu.cn

Xiaoyu Wu*
Communication University of China
State Key Laboratory of Media Convergence and
Communication
Beijing, China
wuxiaoyu@cuc.edu.cn

Shengjin Wang
Tsinghua University
Department of Electronic Engineering
Beijing, China
wgsgj@tsinghua.edu.cn

Jiayao Qian
Communication University of China
State Key Laboratory of Media Convergence and
Communication
Beijing, China
qjy759@cuc.edu.cn

## ABSTRACT

Large-scale pretrained image-language models have shown remarkable performance recently. However, building a video-language model is more challenging due to the complexity of video and the difficulty of collecting high-quality data. This paper builds a video-language model in an adaptive manner, which transfers the knowledge from the image domain and can achieve state-of-the-art performance without any further massive video pretraining. The main contributions include a Visual Perception Adapter that seamlessly and efficiently adapts a pretrained image-language model to the video domain and a fine-grained contrastive learning with Inter-modal Token Alignment that bridges semantic gaps between vision, audio, and language with less data. The proposed model is evaluated on video captioning and retrieval. Experiments demonstrate that the proposed model exhibits competitive performance compared to models pretrained on millions of video-text pairs. Notably, our model's CIDEr and R@1 scores on the MSR-VTT dataset exceed the existing state-of-the-art by 6.3% and 1.3%.

## CCS CONCEPTS

• **Computing methodologies** → **Computer vision tasks**; **Natural language generation**.

## KEYWORDS

Deep learning, Transfer learning, Video captioning, Video retrieval, Multimodality

---

*Corresponding Author

**ACM Reference Format:**
Zihao Liu, Xiaoyu Wu, Shengjin Wang, and Jiayao Qian. 2024. Adaptively Building a Video-language Model for Video Captioning and Retrieval without Massive Video Pretraining. In *Proceedings of the 32nd ACM International Conference on Multimedia (MM '24), October 28–November 1, 2024, Melbourne, VIC, Australia.* ACM, New York, NY, USA, 10 pages. https://doi.org/10.1145/3664647.3680778

## 1 INTRODUCTION

Recently, there has been a growing interest in video-language multimodal learning. Among various tasks, video captioning and video retrieval stand out as two quintessential tasks, demanding models with robust generative and representational capabilities, respectively. While traditional approaches [11, 29, 33, 35, 61] study different video multimodal tasks independently, recent works [7, 23, 25, 48, 60] unify representational and generative tasks through flexible architectures and scale up the model and dataset. While these methods improve the model's generalization, their significant training costs pose challenges for research and application.

As shown in Figure 1, current Vision-Language Models (VLMs) struggle to achieve both low training costs and high generalization. Traditional VLMs [11, 29, 33, 35, 61] are trained on datasets corresponding to different downstream tasks, which are usually manually annotated and high in quality but small in scale. These models lack multi-task capability (including generative and representational tasks) and are insufficiently generalizable. Recently, with the rise of the Big Convergence concept [3, 15], large-scale pretrained VLMs that unify various multimodal tasks have emerged. Some [9, 42] are trained on large-scale video-text datasets [36], achieving fair generalization performance. However, the difficulty of collecting high-quality video-text data results in unsatisfactory performance and incurs high training costs [15]. Some [7, 48, 53] go further by unifying video and image with a flexible architecture, leveraging higher-quality image datasets. They are trained on mixed datasets or through multiple stages, thus achieving excellent performance on multiple downstream tasks. However, the computational costs sharply increase, making training such models a severe problem in scenarios with limited resources. Some methods [30, 31] skip

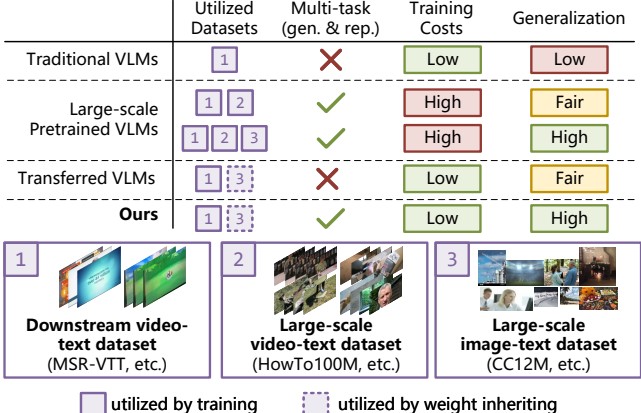

| | Utilized Datasets | Multi-task (gen. & rep.) | Training Costs | Generalization |
|---|---|---|---|---|
| Traditional VLMs | 1 | ✗ | Low | Low |
| Large-scale Pretrained VLMs | 1 2 | ✓ | High | Fair |
| | 1 2 3 | ✓ | High | High |
| Transferred VLMs | 1 3 | ✗ | Low | Fair |
| **Ours** | 1 3 | ✓ | Low | High |

| 1 | 2 | 3 |
|---|---|---|
| **Downstream video-text dataset** (MSR-VTT, etc.) | **Large-scale video-text dataset** (HowTo100M, etc.) | **Large-scale image-text dataset** (CC12M, etc.) |

□ utilized by training     ⬚ utilized by weight inheriting

**Figure 1: Comparison between previous Vision-Language Models (VLMs) and ours. Previous methods utilize multiple datasets to boost generalization of video-language tasks. But they need to trade off between training costs, generalization, and multi-task capacity, whereas our proposed method performs better and has lower training costs.**

heavy pre-training but mostly focus on representational tasks like retrieval or classification. Our method skips video-text pre-training and utilizes an existing image-language model to achieve high generalizability, low training costs and strong multi-task capacity.

To perform multitasking, a unified architecture is crucial. In addition, the strong generalization ability of large-scale pretrained VLMs is largely derived from large-scale image-text datasets, so models trained on such datasets contain valuable knowledge. Employing an image-text pretrained Multiway Transformer [23] as baseline, this paper proposes an adaptive methodology to build a robust, multi-channel, low-training-cost model with satisfactory generalization performance in generative and representational tasks.

We propose two modules to achieve the goal. Firstly, we seamlessly adapt the baseline to the video domain with our proposed **Visual Perception Adapter (VPA)**, which improves the original attention mechanism and enables to model videos with lower costs. Inspired by parameter-efficient fine-tuning (PEFT) methods [19], VPA is designed to adapt the model to data with different distributions while minimizing disturbance to the original structure as much as possible. It is worth noting that most PEFT methods focus on the same tasks and the number of learnable parameters, while ours empower the model to conduct new tasks. VPA comprises a Grouped Temporal Attention (GTA) module and a Patch Feature Dropout (PFD) module. GTA actively reduces complexity by constraining the attention range in cross-attention, which is easy to implement and does not introduce new parameters or alter the entire attention mechanism like other variants with linear complexity [39, 50, 52, 62]. PFD passively reduces complexity by randomly dropping keys/values at each layer. While some works explore token masking during the input stage[17], our PFD works more like dropout, functioning at the feature level and varying at each layer. In addition, to utilize the audio channel in videos, we add audio modality experts to the baseline as in [4], which only increase a small number of Feed Forward Network (FFN) parameters.

Secondly, we propose a fine-grained trimodal contrastive learning with **Inter-modal Token Alignment (ITA)**. To avoid massive video-text pretraining, we directly train the adapted model using high-quality downstream datasets, inevitably facing the semantic gap between images and videos. To address this issue, we better utilize these high-quality data by employing a top-k strategy for finer-grained semantic alignment between different modalities instead of traditional coarse-grained alignment at the sentence/video level. In summary, our contributions are as follows:

- We establish a robust multi-channel video-language model in an adaptive way that maximally leverages the knowledge learned by existing image-language models from large-scale datasets. It seamlessly transfers the knowledge to the video domain without the need for massive video pretraining and is able to perform both representative and generative tasks.
- We propose two components for the adaptive building method: VPA and ITA. VPA optimizes the patch-level spatio-temporal attention mechanism while keeping minor disturbance to the inherited knowledge. ITA establishes token-level fine-grained semantic alignment across three modalities to efficiently utilize downstream datasets.
- Experiments demonstrate the state-of-the-art performance of our model. We take video captioning and retrieval as examples of representative and generative tasks. In particular, our model's CIDEr and R@1 scores on the MSR-VTT dataset exceed the existing state-of-the-art by 6.3% and 1.3%.

## 2 RELATED WORKS

### 2.1 Vision-Language Pretraining

Vision-language pretraining aims at acquiring powerful vision-language representations through large-scale training of noisy image/video-text pairs collected from the Web. These models [21, 25, 40, 60] are usually trained by the contrastive loss or the language modeling loss. BLIP-2 [23] is a state-of-the-art image-language model which combines the advantages of [1, 4, 20, 24] from both architectural and data perspective, however, it cannot handle video tasks. Due to difficulties in utilizing videos effectively, video-text pretraining is more challenging. Videos are usually modeled by temporal encoding modules [27, 43], 3D local attention features [27, 29, 42] or global patch-level self-attention [9]. Furthermore, many video-language models [27, 29, 42] are trained solely on video-text datasets, neglecting the use of high-quality image data, while some also exploit image-text data [9, 48, 53, 56] and achieve better performance. However, these methods incur higher training overhead, and this paper proposes a method that achieves comparable results without extensive video-text pretraining by adapting image models to video tasks.

### 2.2 Adapting Image Models to Video Tasks

Instead of combining image-text data with video-text data for large-scale pretraining, transferring a pretrained image model to the video domain is more efficient. Several approaches improve the effectiveness of video recognition by extending pretrained image models [2, 5, 37, 59]. In addition to these unimodal methods, VideoOFA [9] and VideoCoCa [56] are the closest works to the idea proposed in this paper. They transfer image-text models OFA [49] and CoCa [60],

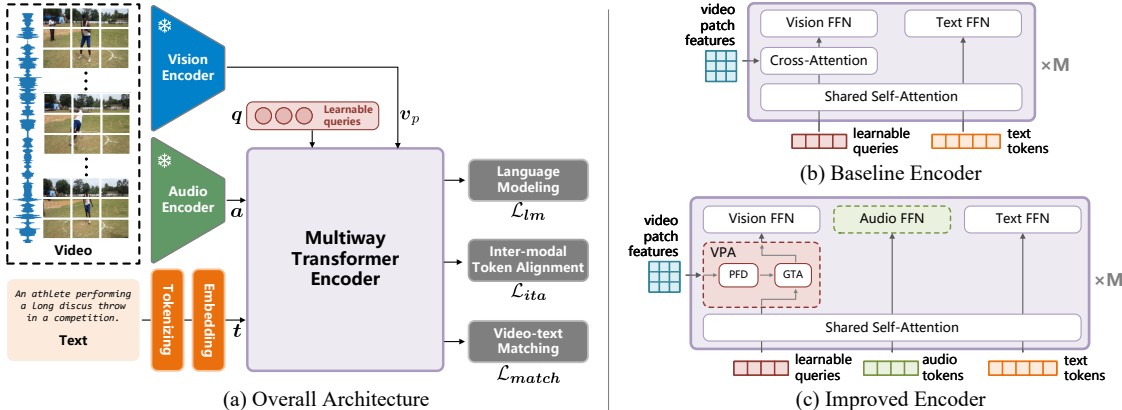

(a) Overall Architecture

(b) Baseline Encoder

(c) Improved Encoder

**Figure 2: Architecture of the proposed model (a) with detail architecture of the baseline encoder (b) [4, 23] and our improved encoder (c). We first extract visual and audio features using pretrained models and then feed them along with text tokens and learnable queries into a Multiway Transformer Encoder. Finally, the model is supervised by multiple losses, including our proposed ITA loss. The baseline encoder consists of a shared self-attention layer, a visual cross-attention layer, and two modality-specific FFNs. Our improved encoder adds a audio pathway and replaces the cross-attention layer with the proposed VPA, which is further composed of PFD and GTA.**

respectively. However, these approaches still require video-text pre-training and suffer from the inefficiency of processing patch-level features. This paper demonstrates that our adapted model achieves better results without further video pretraining. Furthermore, we reduce the computational cost of processing patch-level features through dimension reduction and local attention.

### 2.3 Video Captioning and Video Retrieval

Video captioning and video retrieval are two downstream tasks performed in this paper to validate the adapted model. Most of the previous approaches improve the performance by adopting new network architectures, such as multi-way feature fusion [11, 33, 44], attention mechanisms[11, 58, 61], multimodality[10, 33, 61], special text generation mechanisms[61]. Although these approaches enhance performance in diverse aspects, they are constrained to small-scale datasets with careful labeling. This paper focuses on adapting a pretrained image-language model and transferring knowledge from large-scale datasets to downstream tasks.

## 3 METHOD

Our workflow constructs a Multiway Transformer [4, 23] as baseline network first, followed by adaptive enhancements. This section begins with the constructed overall framework (Figure 2), followed by individual sections on the proposed VPA and ITA modules.

### 3.1 Overall Framework

We start by uniformly sampling $L_v$ frames from the video to form the keyframes set $\mathbf{F}$. Each frame undergoes individual encoding utilizing a pretrained vision encoder ($\mathcal{E}_v$) based on Vision Transformer [14], resulting in patch-level features ($v_p$), supplemented with temporal encoding ($\varepsilon$) [46].

$$v_p^i = \mathcal{E}_v(\mathbf{F}^i) + \varepsilon^i, \quad v_p \in \mathbb{R}^{L_v \times D_v}, \quad (1)$$

where $i$ represents the $i$-th frame in $L_v$ frames and $D_v$ denotes visual feature dimension. The audio is partitioned into $L_a$ non-overlapping segments $\mathbf{A}$. And a Transformer-based pretrained encoder ($\mathcal{E}_a$) [16] and a linear layer is then applied to extract segment-level features ($a$), which are also supplemented with temporal encoding ($\xi^j$).

$$a^j = W_{inp}\mathcal{E}_a(\mathbf{A}^j) + \xi^j, \quad a \in \mathbb{R}^{L_a \times D_m}, \quad (2)$$

where $j$ represents the $j$-th segment in $L_a$ segments, $W_{inp}$ is the learnable weight matrix, and $D_m$ denotes model dimension. For text, we utilize tokenizing and embedding methods identical to [13], obtaining $L_t$ token-level embedding vectors.

$$t = W_{emb}\mathbf{T}, \quad t \in \mathbb{R}^{L_t \times D_m}, \quad (3)$$

where $\mathbf{T}$ is the tokenized one-hot vector of text, and $W_{emb}$ is the learnable embedding matrix. The pretrained visual encoder and audio encoder are kept frozen through the whole process.

Next, these features are fed into a Multiway Transformer encoder [4, 23] comprising $M$ blocks. For simplicity, we will omit the block index and incorporate pre-layer normalization and residual connection within the layer function. Several learnable queries and text tokens are concatenated in a token sequence and fed into a shared self-attention layer. Additionally, by controlling attention mask $\mathbf{m}$, we can either block information interaction between different modalities or implement causal self-attention. Formally,

$$[q^s, t^s] = \Psi_{\theta_s}([q, t], \mathbf{m}), \quad (4)$$

where $\Phi_{\theta_s}$ denotes the self-attention layer. Then, the encoded queries and video patch features are fed into cross-attention to obtain visual information. Furthermore, the outputs of self-attention and cross-attention are encoded by modality-independent FFNs:

$$q^f = \mathcal{F}_{\theta_{fq}}(\Phi_{\theta_c}(q^s, v_p)), \quad t^f = \mathcal{F}_{\theta_{ft}}(t^s), \quad (5)$$

where $\Phi_{\theta_c}$ denotes the cross-attention layer, $\mathcal{F}$ denotes the FFN layer with parameters of different modalities.

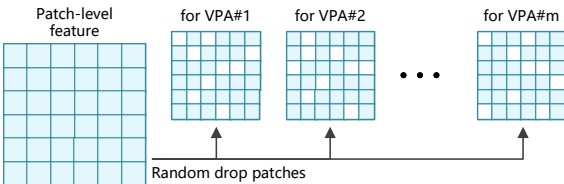

**Figure 3: The structure of PFD. It randomly drops a portion of patch tokens. The dropout results vary across different layers and are performed at feature level.**

Finally, after $M$ blocks of encoding, the outputs of the last block are denoted $[\tilde{q}, \tilde{t}]$ for loss function. The baseline employs three loss functions described in [23], namely, language modeling loss, contrastive loss, and matching loss:

$$\mathcal{L}(\tilde{q}, \tilde{t}) = \mathcal{L}_{lm} + \mathcal{L}_{con} + \mathcal{L}_{match}. \tag{6}$$

These losses cover generative and encoding objectives, which can facilitate both video captioning and retrieval tasks. For more details, we recommend readers to refer to [23].

Our improvements to the baseline are threefold: Firstly, we add input for audio features and its corresponding modality FFN. Secondly, we replaced the cross-attention module with the VPA module, which will be detailed in Section 3.2. Thirdly, we enhanced the contrastive loss to the fine-grained ITA loss, which will be detailed in Section 3.3. It is important to note that the improved model inherits all parameters except for the Audio FFN, thus preventing loss of learned knowledge. Formally, Equations (4)(5)(6) are improved as:

$$[q^s, a^s, t^s] = \Psi_{\theta_s}([q, a, t], \mathbf{m}), \tag{7}$$

$$q^f = \mathcal{F}_{\theta_{fq}}(\Phi'_{\theta_c}(q^s, v_p)), \quad a^f = \mathcal{F}_{\theta_{fa}}(a^s), \quad t^f = \mathcal{F}_{\theta_{ft}}(t^s), \tag{8}$$

$$\mathcal{L}(\tilde{q}, \tilde{a}, \tilde{t}) = \mathcal{L}_{lm} + \lambda_{ita}\mathcal{L}_{ita} + \mathcal{L}_{match}, \tag{9}$$

where $\Phi'_{\theta_c}$ denotes VPA and $\mathcal{L}_{ita}, \lambda_{ita}$ denote ITA loss and its weight.

## 3.2 Visual Perception Adapter

VPA is employed to enable a fix number of learnable vectors to perceive video patch-level features through enhanced cross-attention mechanism while maintaining low complexity. As shown in Figure 2(c), VPA replaces the cross-attention module in the baseline and consists of two parts: PFD and GTA. Video features first undergo random dropout through PFD. Subsequently, they serve as keys/values for grouped local attention with perceptual vectors.

*3.2.1 Patch Feature Dropout.* This module directly reduce the number of patches for efficiency during training. As shown in Figure 3, $P\%$ of patches are randomly dropped before cross-attention:

$$v_p^- = \text{PFD}(v_p) = \begin{pmatrix} v_p^{h_1} & v_p^{h_2} & \cdots & v_p^{h_i} & \cdots & v_p^{h_n} \end{pmatrix} \tag{10}$$

where $h_i$ is the index of the randomly selected patches and $v_p^-$ is the features after PFD, whose length is decreased from $L_v$ to $n = \lfloor L_v \cdot (1 - P\%) \rfloor$. Our method implements this specific dropout at the feature level rather than the pixel level, so that the patch features of the video can be pre-extracted, which also expedites the training process. Moreover, dropping on patch level allows the

**Table 1: Comparison of different attention mechanisms. $T, H, W$ denote dimensions in time, height, and width. $D_m$ denotes feature dimension, $L_q$ denotes the number of queries, and $K, G$ denote model-specific hyperparameters. CA denotes cross-attention.**

| Method | Complexity per Layer | No Extra Params | Applicable to CA |
|---|---|---|---|
| Self-attention[46] | $O(T^2H^2W^2D_m)$ | - | - |
| Cross-attention[46] | $O(THWL_qD_m)$ | ✓ | ✓ |
| LinFormer[50] | $O(KL_qD_m)$ | ✗ | ✓ |
| InFormer[62] | $O(THW \log L_qD_m)$ | ✓ | ✓ |
| CosFormer[39] | $O((THW + L_q)D_m^2)$ | ✓ | ✗ |
| FlowFormer[52] | $O((THW + L_q)D_m^2)$ | ✓ | ✓ |
| **GTA(ours)** | $O(\frac{THW}{G}L_qD_m)$ | ✓ | ✓ |

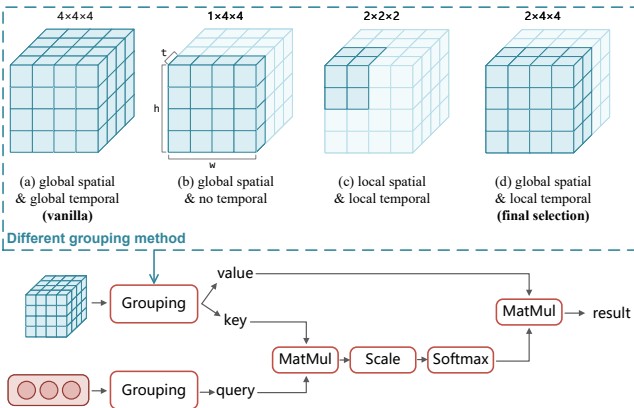

**Figure 4: The structure of GTA. GTA is a variant of cross-attention that actively restricts the attention scope by grouping learnable queries and visual patch features. The size and shape of the groups can be flexibly adjusted.**

model to reason through partial information, facilitating the model's understanding of videos and suppressing overfitting. Compared to BERT [13] or BEiT [3], our method directly drops the tokens instead of replacing them with [mask] tokens, which could lower the computational costs. Compared to MAE [17], our approach drops differently at each layer and refrains from reconstructing the dropped (or masked) tokens, which gets rid of an extra decoder.

*3.2.2 Grouped Temporal Attention.* This module further alleviates the increasing complexity of cross-attention with the growing number of video patches while introducing no extra training parameters. As shown in Table 1, there are various works aiming to reduce the quadratic complexity of self-attention. However, they either involve additional training parameters (row 3), offer limited reduction in complexity (row 4), cannot be applied to cross-attention (row 5), or significantly changes the original attention calculation (row 6). It is crucial to note that such works mainly focus on scaling up language models or handling long sequence time-series data, which differs from the case in this paper.

Our approach targets a visual perception mechanism with learnable vectors as queries and video patch features as keys/values. Therefore, prior methods like CosFormer[39], which enhances the weights of the diagonal region in the attention matrix, are not applicable. However, our situation allows for the introduction of a different prior knowledge: the high correlation between spatially and temporally adjacent patches in videos. This correlation enables the extraction of local structures and motion information, sharing similarities with the benefits of using CNNs to process images.

Specifically, we first reshape $v_p$ of dimensions $L_v \times D_v$ into a tensor of dimensions $T \times H \times W \times D_v$, which can be viewed as a cube where each small block is a vector of dimension $D_v$. Then, as shown in Figure 4, this cube is divided into $G$ non-overlapping windows of size $t \times h \times w$, and queries ($q$) are also divided into $G$ groups, each with a length of $l_q$. The size and shape of the windows, as well as the number of queries, can be adjusted arbitrarily while ensuring that $t, h, w$ are positives. The upper part of Figure 4 shows a couple of examples. Next, queries and the windowed patches from the same group perform cross-attention with a time complexity of $O(thwl_q D_m) = O(THWL_q D_m / G^2)$:

$$\hat{q}_g = \text{softmax}(\frac{W_q \mathcal{G}(q)_g \mathcal{G}(v_p)_g^\top W_k^\top}{\sqrt{D_m}})W_v \mathcal{G}(v_p)_g^\top, \quad (11)$$

where $\mathcal{G}(\cdot)_g$ denotes the grouping operation as described above, the subscript $g$ denotes the $g$-th group, and $W_q, W_k, W_v$ are learnable matrices. Equation (11) is repeated $G$ times, which is implemented by batching and results in a complexity of $O(\frac{THW}{G}L_q D_m)$ for GTA.

In Figure 4, Example (a) represents the vanilla approach of full attention (see Figure 2(b)). This method applies global spatiotemporal attention, exhibiting high complexity and the possibility of introducing noise. Example (b) illustrates the case with a window size of $1 \times 4 \times 4$, involving global spatial attention and local temporal attention modeling, similar to the approach in [23] for image processing. Example (c) depicts the scenario of $2 \times 2 \times 2$, incorporating local spatiotemporal attention, allowing each query to focus on a small spatiotemporal region. Example (d) showcases the situation a of $2 \times 4 \times 4$ window, featuring global spatial attention and local temporal attention modeling. This aligns with the pretrained model we used, mitigating the discrepancy between pretraining and downstream tasks. Additionally, this method facilitates modeling of short-term temporal information, avoiding introducing significant noise when directly modeling long-term temporal information.

To accelerate training using mini-batches and when simultaneously using GTA and PFD, it is necessary to keep the number of dropped patches within each window to be the same. Therefore, Equation (11) is rewritten as:

$$\hat{q}_g = \text{softmax}(\frac{W_q \mathcal{G}(q)_g \mathcal{G}(v_p)_g^{-\top} W_k^\top}{\sqrt{D_m}})W_v \mathcal{G}(v_p)_g^{-\top}, \quad (12)$$

where the superscript minus symbol denotes the PFD operation as in Equation (10). Therefore the $\Phi'_{\theta_c}$ in Equation (8) is rewritten as:

$$\Phi'_{\theta_c}(q^s, v_p) = \text{Concat}(\hat{q}_1; \hat{q}_2; \cdots; \hat{q}_G), \quad (13)$$

where $\text{Concat}(\cdot)$ denotes concatenation. As shown in Figure 2, the $[q^f, a^f, t^f]$ of the last block are denoted as $[\tilde{q}, \tilde{a}, \tilde{t}]$.

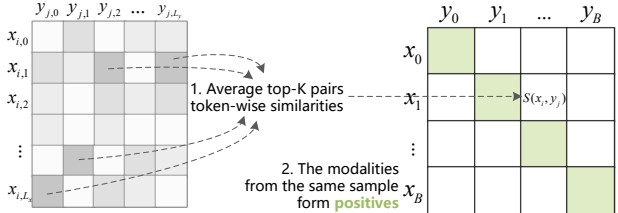

**Figure 5: The process of the trimodal contrastive loss with inter-modal token alignment.**

## 3.3 Trimodal Contrastive Learning with Inter-modal Token Alignment

Contrastive learning has been demonstrated in a number of works to be beneficial in bridging the semantic gap between different modalities [7, 23, 40, 58]. To enable the adapted model to fully exploit the information from multichannel videos, we propose a fine-grained vision-audio-text contrastive loss to align multimodal features. The vanilla contrastive loss based on InfoNCE [45] is as follows:

$$\mathcal{L}_{NCE}(x, y, \mathcal{N}_y) = -\log(\frac{e^{S(x,y)/\eta}}{e^{S(x,y)/\eta} + \sum_{y' \in \mathcal{N}_y} e^{S(x,y')/\eta}}), \quad (14)$$

where $x$ and $y$ are positive pairs from two modalities (e.g., visual queries and audio tokens), $\mathcal{N}_y$ is the set of negative samples of $y$ modality from the rest of the batch, $\eta$ is the temperature coefficient, and $S(\cdot, \cdot)$ is the similarity function, typically using cosine similarity.

In the context of this paper, both $x$ and $y$ are feature sequences. To obtain the similarity between them, common approaches include using pooling or additional encoding layers [34] to get the overall features of the sequences, followed by calculating cosine similarity. We argue that such methods either fail to utilize fine-grained relations between modalities or introduce additional parameters. Another model [23], which is also based on the Perceiver architecture [20], computes the similarity between each query and the text's [CLS] feature, and then selects the maximum value as the final similarity. However, in our case, each query only represents local visual information from the video, and combining information from multiple queries is necessary for better alignment with the text. Therefore, we generalize the calculation of $S$ using a top-K strategy as illustrated in Figure 5 and following equations:

$$S_{xy} = \frac{xy^\top}{||x||_2 ||y||_2}, \quad S(x, y) = \sum_{s \in \text{TopK}(S_{xy})} \frac{s}{K}, \quad (15)$$

where $\text{TopK}(\cdot)$ indicates taking the largest $K$ value from the matrix. The outputs of the last block of the backbone are denoted $[\tilde{q}; \tilde{a}; \tilde{t}]$, and we calculate the bidirectional contrastive losses between these three modalities:

$$\mathcal{L}_{ita}^{qa} = \frac{1}{2}(\mathcal{L}_{NCE}(\tilde{q}, \tilde{a}, \mathcal{N}_a) + \mathcal{L}_{NCE}(\tilde{a}, \tilde{q}, \mathcal{N}_q)), \quad (16)$$

$$\mathcal{L}_{ita}^{qt} = \frac{1}{2}(\mathcal{L}_{NCE}(\tilde{q}, \tilde{t}, \mathcal{N}_t) + \mathcal{L}_{NCE}(\tilde{t}, \tilde{q}, \mathcal{N}_q)), \quad (17)$$

$$\mathcal{L}_{ita}^{at} = \frac{1}{2}(\mathcal{L}_{NCE}(\tilde{a}, \tilde{t}, \mathcal{N}_t) + \mathcal{L}_{NCE}(\tilde{t}, \tilde{a}, \mathcal{N}_a)), \quad (18)$$

**Table 2: Comparison with state-of-the-art methods on video captioning. PT denotes pretraining. M, A, and O in *Extra Modalities* respectively denotes motion, audio, and others. Blue indicates the highest metrics of large models and large-scale pretrained models. Red indicates the highest metrics of the other models. † indicates methods optimized by SCST[41].**

| Method | Vision Encoder | PT Video-Text Data | Learnable Params | Extra Modalities | MSR-VTT | | | | VATEX | | | | MSVD | | | |
|---|---|---|---|---|---|---|---|---|---|---|---|---|---|---|---|---|
| | | | | | B@4 | M | R | C | B@4 | M | R | C | B@4 | M | R | C |
| *Methods with large model (> 1B) or large-scale video-text pretraining (> 10M)* | | | | | | | | | | | | | | | | |
| CLIP4Cap[43] | $CLIP_B$ | 136M | - | - | 47.2 | 31.2 | 64.8 | 60.0 | 40.6 | - | 54.5 | 85.7 | - | - | - | - |
| MV-GPT[42] | $ViViT_B$ | 136M | - | O | 48.9 | 29.9 | 64.0 | 60.0 | - | - | - | - | - | - | - | - |
| VideoCoCa[60] | ViT-40L | 8.7M | 1.1B | - | 53.8 | - | 68.0 | 73.2 | 39.7 | - | 54.5 | 77.8 | - | - | - | - |
| $VALOR_L^†$[7] | $CLIP_L$ | 13.5M | 593M | A | 54.4 | 32.9 | 68.0 | 74.0 | 45.6 | 29.4 | 57.4 | 95.8 | 80.7 | 51.0 | 87.9 | 178.5 |
| $GIT2^†$[48] | DaViT | 0M | 5.1B | - | 54.8 | 33.1 | 68.2 | 75.9 | 42.7 | 28.8 | 56.5 | 94.5 | 82.2 | 52.3 | 88.7 | 185.4 |
| $VAST^†$[8] | $CLIP_G$ | 297M | 1.3B | A | 56.7 | - | - | 78.0 | 45.0 | - | - | 99.5 | - | - | - | - |
| *Methods with video-text pretraining* | | | | | | | | | | | | | | | | |
| mPLUG-2-base†[53] | $CLIP_B$ | 2.5M | 618M | - | 52.2 | 32.1 | 66.9 | 72.4 | - | - | - | - | 69.3 | 45.1 | 81.9 | 148.2 |
| mPLUG-2† | $CLIP_L$ | 2.5M | 900M | - | 57.8 | 34.9 | 70.1 | 80.3 | - | - | - | - | 75.0 | 48.4 | 85.3 | 165.8 |
| VideoOFA†[9] | ResNet101 | 2.2M | 400M | - | 50.5 | 33.1 | 66.8 | 73.5 | 39.6 | 27.2 | 54.2 | 79.5 | 75.9 | 47.7 | 85.0 | 165.5 |
| $VALOR_B^{-†}$ | $VidSwin_B$ | 2.5M | 342M | A | 48.1 | 30.4 | 64.3 | 61.5 | 40.7 | 26.1 | 53.8 | 71.6 | 74.3 | 47.1 | 83.8 | 156.1 |
| $VALOR_B^†$ | $VidSwin_B$ | 3.5M | 342M | A | 53.8 | 32.3 | 67.0 | 66.6 | 41.9 | 26.6 | 54.6 | 73.9 | 76.1 | 48.0 | 85.2 | 162.1 |
| *Methods without video-text pretraining* | | | | | | | | | | | | | | | | |
| OpenBook[61] | IncepV2 | 0M | - | M+O | 33.9 | 23.7 | 50.2 | 52.9 | 33.9 | 23.7 | 50.2 | 57.5 | - | - | - | - |
| SwinBERT[29] | $VidSwin_B$ | 0M | 230M | - | 41.9 | 29.9 | 62.1 | 53.8 | 38.7 | 26.2 | 53.2 | 73.0 | 58.2 | 41.3 | 77.5 | 120.6 |
| $GIT_B^†$ | $CLIP_B$ | 0M | 129M | - | 46.6 | 29.6 | 63.2 | 57.8 | 37.9 | 24.4 | 51.9 | 60.0 | 69.3 | 44.5 | 81.4 | 142.6 |
| $GIT_L^†$ | $CLIP_L$ | 0M | 347M | - | 48.7 | 30.9 | 64.9 | 64.1 | 41.6 | 26.2 | 54.3 | 72.5 | 75.8 | 48.7 | 85.5 | 162.9 |
| $GIT^†$ | CoSwin | 0M | 681M | - | 53.8 | 32.9 | 67.7 | 73.9 | 41.6 | 28.1 | 55.4 | 91.5 | 79.5 | 51.1 | 87.3 | 180.2 |
| **Ours**(unimodal)† | $CLIP_L$ | 0M | 183M | - | 51.5 | 33.6 | 67.8 | 78.5 | 43.1 | 28.2 | 55.9 | 86.8 | 72.9 | 48.6 | 85.3 | 169.9 |
| **Ours**† | $CLIP_L$ | 0M | 240M | A | 53.5 | 34.3 | 69.0 | 80.7 | 44.1 | 28.5 | 56.5 | 88.0 | - | - | - | - |

*Note:* The grey item of MV-GPT is re-evaluated by [57]. The grey parameters count of mPLUG-2-base is estimated by us, and the grey results on MSR-VTT indicate they use a different split, leading to an unfair comparison.

$$\mathcal{L}_{ita} = \frac{1}{3}(\mathcal{L}_{ita}^{qa} + \mathcal{L}_{ita}^{qt} + \mathcal{L}_{ita}^{at}). \quad (19)$$

16 frames for DiDeMo dataset. Please refer to the supplementary material for detailed experimental setups.

## 4 EXPERIMENTS

### 4.1 Experiment setup

We perform a comprehensive evaluation of our proposed method on six benchmarks covering two downstream tasks: video captioning and video retrieval.

For video captioning, we evaluate on three widely used datasets (MSVD [6], MSR-VTT [54], VATEX [51]), as well as one novel dataset (VALOR-32k [7]). We utilize the following metrics to evaluate the performance of our model: Bleu@4 (B@4) [38], METEOR (M) [12], ROUGE-L (R) [28], and CIDEr (C) [47]. During the inference process, we employ beam search with a beam size of 3, which is generally used in many video captioning methods. Video retrieval performance is evaluated using the MSR-VTT dataset [54] and the DiDeMo dataset [18]. We use text-to-video recall at rank K (R@K, where K=1, 5, 10) as our evaluation metrics. And we follow the same evaluation process as in [23]. We use CLIP-ViT-L/14 [40] and AST [16] finetuned on AudioSet as the visual encoder and the audio encoder. We also inherit the weights of BLIP-2 [23]. By default we empirically extract 8 frames at equal intervals for datasets (MSR-VTT, MSVD, VATEX, VALOR) with shorter video lengths, and

### 4.2 Comparison with SOTA methods

We compare the proposed model with other state-of-the-art models in various aspects. As shown in Figure 6, our model achieves leading results both in trainable parameters and the video-text pairs used, demonstrating the superiority of our adaptive approach. In detail, as shown in Table 2 and Table 3, compared to models without video-text pretraining, our model basically achieves state-of-the-art results on all five benchmarks. One exception is the GIT [48] on video captioning, which utilizes up to 800M higher-resolution image-text pairs for pretraining. In the captioning task, our model outperforms methods with parameters below 1B and training pairs below 10M. Compared to methods with higher training cost, our approach still outperforms methods with pretraining data exceeding 100M (CLIP4Cap [43], MV-GPT [42]) and the model with 5 times the parameters (VideoCoCa [56]). It even outperforms methods with training parameters more than 20 times larger than ours (GIT2 [48]) on the MSR-VTT dataset. It's important to note that the GIT2 model, which achieved the best comprehensive metrics, constructs a model with 5.1B parameters and uses up to 12.9B image-text pairs for training, and thus exhibits a significant improvement compared to

Table 3: Comparison with state-of-the-art methods on video retrieval. PT denotes pretraining. M, A, and O in *Extra Modalities* respectively denotes motion, audio, and others. Blue indicates the highest metrics of video-text pretrained methods. Red indicates the highest metrics of others. † indicates methods applying DSL [11].

| Method | Vision Encoder | PT Video-Text | Learnable Params | Extra Modalities | MSR-VTT | | | DiDeMo | | |
|---|---|---|---|---|---|---|---|---|---|---|
| | | | | | R@1 | R@5 | R@10 | R@1 | R@5 | R@10 |
| *Methods with video-text pretraining* | | | | | | | | | | |
| VALOR$_B$[7] | VidSwin$_B$ | 3.5M | 342M | A | 43.0 | 72.2 | 82.1 | 52.2 | 80.8 | 86.8 |
| CLIP4Clip[34] | CLIP$_B$ | 136M | - | | 44.5 | 71.4 | 70.1 | 43.4 | 70.2 | 80.6 |
| mPLUG-2[53] | CLIP$_L$ | 2.5M | 900M | - | 53.1 | 77.6 | 84.7 | 56.4 | 79.1 | 85.2 |
| CLIP-ViP$^†$[55] | CLIP$_B$ | 100M | - | O | 57.7 | 80.5 | 88.2 | 55.3 | 82.0 | 89.3 |
| UMT[26] | CLIP$_L$ | 12.5M | 304M | - | 58.8 | 81.0 | 87.1 | 70.4 | 90.1 | 93.5 |
| VALOR$_L^†$ | CLIP$_L$ | 13.5M | 593M | A | 59.9 | 83.5 | 89.6 | 61.5 | 85.3 | 90.4 |
| VAST[8] | CLIP$_G$ | 297M | 1300M | A | 63.9 | 84.3 | 89.6 | 72.0 | 89.0 | 91.4 |
| *Methods without video-text pretraining* | | | | | | | | | | |
| CAMoE$^†$[11] | CLIP$_B$ | 0M | - | - | 47.3 | 74.2 | 84.5 | 43.8 | 71.4 | - |
| X-CLIP[35] | CLIP$_B$ | 0M | - | - | 49.3 | 75.8 | 84.8 | 47.8 | 79.3 | - |
| HCMI-L[22] | CLIP$_L$ | 0M | - | - | 49.5 | 74.2 | 83.9 | 41.8 | 71.2 | 79.0 |
| TABLE[10] | CLIP$_B$ | 0M | - | M+A+O | 52.3 | 78.4 | 85.2 | 51.8 | 77.5 | 85.1 |
| STAN$^†$[32] | CLIP$_B$ | 0M | - | - | 54.1 | 79.5 | 87.8 | 54.6 | 78.4 | 85.1 |
| **Ours** | CLIP$_L$ | 0M | 240M | A | 59.4 | 80.8 | 87.4 | 62.1 | 82.3 | 83.3 |
| **Ours**$^†$ | CLIP$_L$ | 0M | 240M | A | 60.7 | 81.9 | 87.0 | 65.3 | 82.6 | 85.1 |

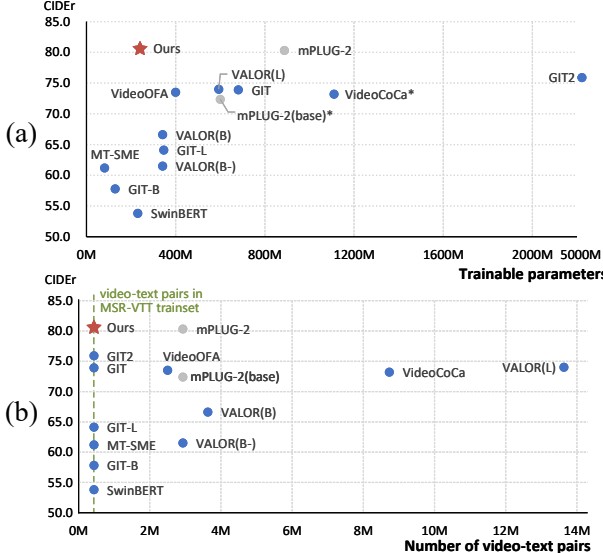

Figure 6: Comparison on MSR-VTT dataset with other methods in terms of (a) trainable parameters and (b) the number of video-text pairs used.

all other methods. In contrast, our method focuses on constructing a model with reasonable parameters, leveraging existing visual-language models, and achieving superior performance without additional video-text pretraining. In the retrieval task, compared to methods with video-text pretraining, our approach achieves competitive results, surpassing CLIP4Clip [34] and CLIP-ViP [55], which utilize over 100 million video-text pairs. Additionally, our

Table 4: Comparison of VALOR dataset on video captioning. Red and Blue indicate the optimal and sub-optimal metrics.

| Method | Params | PT | Audio | B@4 | M | R | C |
|---|---|---|---|---|---|---|---|
| SwinBERT[29] | 230M | 0M | ✗ | 5.4 | 10.7 | 27.2 | 27.3 |
| VALOR$_B^-$[7] | 342M | 2.5M | ✗ | 8.0 | 13.5 | 29.4 | 44.3 |
| VALOR$_B$ | 342M | 3.5M | ✓ | 8.9 | 14.8 | 30.8 | 55.7 |
| VALOR$_L$ | 593M | 13.5M | ✓ | 9.6 | 15.4 | 31.8 | 61.5 |
| VAST[8] | 1300M | 297M | ✓ | 9.9 | - | - | 62.2 |
| **Ours** | 240M | 0M | ✓ | 9.6 | 15.4 | 30.8 | 52.1 |

model achieves comparable performance to UMT [26] and VALOR$_L$ [7], which employed video datasets exceeding 10 million instances. For models with audio, we surpass them with R@1 scores of 60.7 and 65.3 in retrieval and a CIDEr score of 80.7 in captioning. We also achieve the best results in B@4 and METEOR metrics on the VALOR dataset, which has a higher demand for audio understanding. It is important to note that VALOR and VAST are trained on much larger and audio-focused corpus. However, our method achieves competitive results with much less audio for training. VideoOFA [9] and VideoCoCa [56] are the closest works to our approach, both aiming at enhancing video-language learning using existing image-language models. Compared to them, our model has fewer trainable parameters and achieves better results without millions of videos for pretraining.

## 4.3 Ablation Study

*4.3.1 Effectiveness of Each Component.* As shown in Table 5, we perform ablations experiments on three datasets and two tasks by

**Table 5: Ablation on each component on three datasets.**

| Audio | GTA | PFD | $\mathcal{L}_{ita}$ | MSR-VTT CIDEr | VALOR CIDEr | DiDeMo R@1 |
|---|---|---|---|---|---|---|
|  |  |  |  | 72.6 | 43.0 | 55.7 |
| ✓ |  |  |  | 73.1 | 45.3 | 60.0 |
| ✓ | ✓ |  |  | 75.1 | 49.1 | 60.5 |
| ✓ |  | ✓ |  | 73.8 | 50.4 | 60.1 |
| ✓ | ✓ | ✓ |  | 74.9 | 49.3 | 60.7 |
| ✓ |  |  | ✓ | 75.2 | 49.9 | 60.7 |
| ✓ | ✓ |  | ✓ | 76.6 | 50.6 | 61.4 |
| ✓ |  | ✓ | ✓ | 76.7 | 51.3 | 60.8 |
| ✓ | ✓ | ✓ | ✓ | **77.5** | **52.2** | **62.1** |

**Table 6: Comparison between GTA and some linear-complexity attention mechanisms.**

| Method | Complexity | VALOR CIDEr |
|---|---|---|
| Vanilla CrossAttention | $O(THWL_qD_m)$ | 45.3 |
| LinFormer[50] | $O(KL_qD_m)$ | 44.8 |
| InFormer[62] | $O(THW\log L_qD_m)$ | 50.0 |
| FlowFormer[52] | $O((THW+L_q)D_m^2)$ | 45.9 |
| **GTA(ours)** | $O(\frac{THW}{G}L_qD_m)$ | **52.2** |

gradually adding modules to the baseline (Figure 2(b) and Equation (6)). We found that the audio modality, GTA module, PFD module, and ITA loss all effectively enhance the model's performance. The best results are obtained by employing all four strategies. More ablations of GTA and ITA are in Section C of the supplementary.

*4.3.2 Comparison between GTA and Linear-complexity Attention Mechanisms.* We compare several linear-complexity attention mechanisms mentioned in Table 1, and the results are presented in Table 6. We replace the cross-attention with their improved attention mechanism according to their open-source codes. We found that introducing new learnable parameters (LinFormer [50]) or changing the operation of attention mechanism (FlowFormer [52]) can disrupt the model's original knowledge, leading to unsatisfactory performance. InFormer [62] performs slightly better than others, but in the context of this paper, limiting the length of the query has minimal impact on efficiency improvement. Our proposed GTA involves minimal modifications to cross-attention and is tailored for scenarios that require adapting an image-text pretrained cross-attention to the video domain while minimizing disruption to features.

*4.3.3 Efficiency Analysis.* The proposed model's GPU processing speed and peak memory consumption during training are measured for analysis efficiency. As shown in Table 7, the model achieves a time saving of 50.5% and a memory saving of 36.9% with a dropout rate of 50%. Restricting attention to adjacent two sampled frames saved 31.7% of the time, but due to the creation of intermediate variables, there is a slight increase in memory consumption. Combining both strategies results in a slight decrease in model performance.

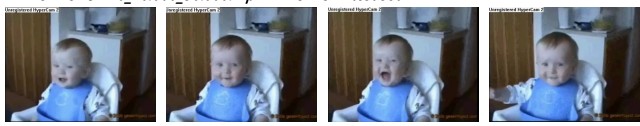

- *Y9YhG2CFTz0_20.000_30.000.mp4 VALOR-32k test set*

Ground Truth:  In the room a child in a blue bib was laughing, and the boy laughed more as the man spoke.
**SwinBERT:** a baby is sitting in a high chair and laughing
**GIT-Large:** a baby is sitting in a high chair and smiling
**Ours:** in the room, a baby in a blue bib sat in a white chair laughing

**Figure 7: Qualitative analysis on video captioning.**

**Table 7: Ablation experiments of PFD and GTA on VALOR dataset. The *relative* columns indicate the relative ratio of the other experiments compared to the first row.**

| Ratio | Window | Time | | GPU Memory | | CIDEr |
|---|---|---|---|---|---|---|
| | | ms | relative | MB | relative | |
| 0% | - | 5082 | 1.000× | 36730 | 1.000× | 43.3 |
| 10% | - | 4653 | 0.916× | 34011 | 0.926× | 44.5 |
| 30% | - | 2949 | 0.580× | 28587 | 0.778× | 44.8 |
| 50% | - | 2515 | 0.495× | 23159 | 0.631× | 45.2 |
| 70% | - | 1344 | 0.264× | 17736 | 0.483× | 44.4 |
| 0% | 32x2x16x16 | 3471 | 0.683× | 38995 | 1.062× | 45.1 |
| 50% | 64x2x16x16 | 2160 | 0.425× | 28834 | 0.785× | 44.3 |
| 30% | 64x2x16x16 | 2813 | 0.554× | 32380 | 0.882× | **45.8** |

We attribute this to an excessive restriction on the model. The best results are obtained when the dropout rate was reduced to 30%.

*4.3.4 Visualization.* Figure 7 is a sample from the dataset. In this example, where other models predict similar simple descriptions, our model predicts more fine-grained descriptions, which we attribute to our fine-grained features and ITA loss function. More qualitative results can be found in the supplementary.

## 5 CONCLUSIONS

This paper presents a novel and efficient approach to adaptively construct a video-language model. The proposed method transfers knowledge from the image-language domain to the video domain, efficiently processes the abundant information in videos through a Visual Perception Adapter, and narrows the semantic gaps between vision, audio, and text modalities by employing a fine-grained tri-modal contrastive learning with Inter-modal Token Alignment. Extensive experiments validate that the constructed model attains satisfactory performance without the need for large-scale video-text pretraining. **Limitation**: Our model focuses on processing short videos and is not able to handle longer videos. To address this issue, we plan to explore efficient and low-data-demand methods for modeling longer videos. **Broad Impact**: The outstanding performance of this model demonstrates its potential to be applied to multimodal large language models, where it can serve as an efficient video feature extractor.

## ACKNOWLEDGMENTS

The authors would like to thank Dr. Linlin Yang for discussions. This work was supported by the state key development program in 14th Five-Year under Grant No. 2021YFF0900701, 2021YFF0602103, 2021YFF0602102, 2021QY1702, and in part by Natural Science Foundation of China (No.61801441). We also thank the research funds under Grant No. 2019GQG0001 from the Institute for Guo Qiang, Tsinghua University, and the High-quality and Cutting-edge Disciplines Construction Project for Universities in Beijing (Internet Information, Communication University of China).

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
