# OpenReview forum: "Adaptively Building a Video-language Model for Video Captioning and Retrieval without Massive Video Pretraining"
_acmmm.org/ACMMM/2024/Conference — MM2024 Poster_

### Official Review · Reviewer_Nr3g · 2024-05-24

**Rating:** 6
**Confidence:** 3

**Summary:**

The paper introduces a framework for adapting an image-language model for video-language tasks. The methods consist of a novel Visual Perception Adapter that encodes videos at a lower cost, and a fine-grained trimodal contrastive learning with Inter-modal Token Alignment. The paper includes extensive experiments and analyses to support the effectiveness of the proposed methods.

**Strengths:**

- The paper is easy to follow.
- The proposed methods are novel and effective in adapting an image-text model for video-language tasks.
- The experiments are extensive.

**Limitations:**

- Adding some Video QA tasks can make the experiments more solid.

- Some suggestions:

1) There is insufficient information on ITA in the introduction section.
2) Figure 1 could be considered for placement on the first page.
3) Figure 2 might be improved by including some brief information about modules like PFD, GTA, ITA, etc., and emphasizing the design of the method in the main figure.

**Suitability:**

3

---

### Official Review · Reviewer_7ofk · 2024-05-25

**Rating:** 4
**Confidence:** 4

**Summary:**

This paper presents an innovative approach to adapting an image-text pretrained model, BLIP-2, into an efficient video-text model. The resulting model demonstrates impressive performance on both video captioning and video retrieval tasks across various benchmarks, achieving these results without the need for extensive video-text pretraining. The authors highlight the high performance and reduced training costs associated with their method.

**Strengths:**

1. The paper is clearly written, well-structured, and easy to follow.

2. The experiments are thorough, demonstrating the proposed model's competitive performance in both video captioning and retrieval tasks without relying on video-text pretraining.

3. The Visual Perception Adapter (VPA) is a novel contribution. Ablation studies validate its effectiveness in reducing computational costs while maintaining high performance in video modeling.

**Limitations:**

1. Further Potential with Video-Text Pretraining: Although the authors emphasize that their method does not require video-text pretraining, it raises curiosity about whether incorporating video-text pretraining could further enhance the model's performance.

2. Dependence on Image-Text Pretrained Model: The method relies on a pretrained image-text model as the foundation. There is a lack of discussion on how the choice of this pretrained model and the effectiveness of the image-text pretraining might influence the proposed method's performance. Currently, experiments are conducted using only a single baseline, BLIP2.

3. Insufficient Citation and Discussion of Stronger SOTA Methods: The paper lacks sufficient references and discussions of stronger SOTA methods, such as "VAST: A Vision-Audio-Subtitle-Text Omni-Modality Foundation Model and Dataset" (NeurIPS 2024). Including such comparisons would provide a more comprehensive understanding of the model's relative performance.

**Suitability:**

3

---

### Official Review · Reviewer_7pfo · 2024-05-28

**Rating:** 3
**Confidence:** 4

**Summary:**

Recently, large-scale pretrained image-language models have shown remarkable performance recently. However, how to train a strong video-language model under training costs is more challenging due to the complexity of video and the difficulty of collecting high-quality data. This paper proposes two modules called VPA and ITA to build a video-language model in an adaptive manner, which transfers the knowledge from the image domain and can achieve state-of-the-art performance without any further massive video pretraining. However, there are still some issues in the paper that need to be addressed.

**Strengths:**

1.	The pipeline proposed in this paper is particularly clear and easy to understand, which makes it accessible even to those who may not be experts in the field. This straightforward approach ensures that researchers can understand and apply the methods discussed.
2.	The paper writing is commendable for its clarity and straightforwardness. The authors use simple and direct language, making complex concepts more understandable.
3.	The paper conducts sufficient experiments to verify the effectiveness of the proposed method for video captioning and retrieval.
4.	The proposed method has the potential for further improvement.

**Limitations:**

1.	Performing Parameter-Efficient Fine-Tuning (PEFT) on downstream tasks based on a general large model is a common approach. However, you should compare your proposed VPA with some mainstream PEFT methods to demonstrate its advantages, such as LoRA, adapters, soft prompts, and so on.
2.	Can the method you proposed be used in other downstream tasks, such as video question answering?  Other works, such as mPLUG-2, have been validated on more downstream tasks.
3. Your study improves upon the Q-former of BLIP-2, incorporating new components. However, as demonstrated in Table 5, the proposed GTA and PFD do not markedly boost performance. Does this mean that even without using GTA and PFD, the performance is already quite good with the original setup?
4. If addressing the aforementioned issues, it would greatly benefit your work.

**Suitability:**

3

---

### Official Review · Reviewer_n6JY · 2024-06-07

**Rating:** 3
**Confidence:** 4

**Summary:**

This paper addresses the challenges of building video-language models by transferring knowledge from large-scale pretrained image-language models, thus avoiding the need for extensive video pretraining. The key contributions include a Visual Perception Adapter that adapts image-language models to the video domain efficiently and a fine-grained contrastive learning approach with Inter-modal Token Alignment to bridge semantic gaps between vision, audio, and language. The model is evaluated on video captioning and retrieval tasks, demonstrating competitive performance compared to models pretrained on millions of video-text pairs.

**Strengths:**

1. The tackled problem is important.
2. The proposed method is intuitive.
3. The paper is overall easy to follow.

**Limitations:**

1. The current manuscript over-claims the difference. For example, in Figure 1, the message audience probably receives is that this is the first work leverages existing pretrained model to avoid pretraining on video. But there has been an existing extensive effort along this line [a,b,c]. However, the current manuscript fails to discuss and acknowledge these existing work, and thus leave the audience a wrong impression about the position of the work. It is better to compare with at least a few of these work on multi-channel inputs, which were also shown to achieve SOTA performance [a,b,c] when without any pretraining on videos.

2. The authors stressed on the generalization but there is no evidence on any common ways of showing generalization, such as cross-dataset evaluation or out of distribution evaluation.

[a] VX2TEXT: End-to-End Learning of Video-Based Text Generation From Multimodal Inputs

[b] Socratic Models: Composing Zero-Shot Multimodal Reasoning with Language

[c] Towards Fast Adaptation of Pretrained Contrastive Models for Multi-channel Video-Language Retrieval

**Suitability:**

3

---

### Meta-Review · Area_Chair_yFwD · 2024-07-07

**Recommendation:** Accept (Poster)
**Confidence:** 4

**Metareview:**

Initially, the paper received mixed reviews, with ratings oscillating between "Borderline Reject" and "Borderline Accept." Reviewers appreciated the clarity of the presentation and the novelty of the proposed methods but raised concerns about the over-claims regarding the novelty, lack of sufficient comparison with state-of-the-art methods, and generalization evidence.

After the rebuttal, the authors addressed several concerns, particularly around the novelty and comparison with existing works. However, some issues such as the generalization of the model across different datasets and tasks, and the dependence on the choice of the pretrained image-text model remained partially addressed. The final ratings leaned more towards acceptance after considering the rebuttal and additional clarifications provided by the authors.

---

### Meta-Review · Senior_Area_Chairs · 2024-07-10

**Recommendation:** Accept (Poster)
**Confidence:** 4

**Metareview:**

This paper received mixed ratings initially. After rebuttal, two reviewers who gave BR ratings did not submit the final ratings. AC carefully read the reviews and rebuttal and tend to accept the paper. SAC agrees with AC and recommend acceptance of the paper.